# LC-QTOF/MS-Based Profiling of the Phytochemicals in Ice Plant (*Mesembryanthemum crystallinum*) and Their Bioactivities

**DOI:** 10.3390/foods13121820

**Published:** 2024-06-10

**Authors:** Mira Oh, Ah-Ram Han, Jaeyoun Lee, Sang Yoon Choi, Jae Woong Choi, Nho-Eul Song, Hee-Do Hong, Young Kyoung Rhee, Chang-Won Cho

**Affiliations:** 1Traditional Food Research Group, Korea Food Research Institute, Wanju-gun 55365, Republic of Korea; o.mira@kfri.re.kr (M.O.); lambo3@kfri.re.kr (A.-R.H.); choijw@kfri.re.kr (J.W.C.); nesong@kfri.re.kr (N.-E.S.); honghd@kfri.re.kr (H.-D.H.); ykrhee@kfri.re.kr (Y.K.R.); 2College of Pharmacy, Yonsei Institute of Pharmaceutical Sciences, Yonsei University, Incheon 21983, Republic of Korea; jaeyoun1024@yonsei.ac.kr; 3Functional Food Materials Research Group, Korea Food Research Institute, Wanju-gun 55365, Republic of Korea; sychoi@kfri.re.kr

**Keywords:** phytochemicals, anti-inflammatory, plant extracts, molecular network, bioactive compounds

## Abstract

Recent assessments of the correlations between food and medicine underscore the importance of functional foods in disease prevention and management. Functional foods offer health benefits beyond basic nutrition, with fresh fruits and vegetables being particularly prominent because of their rich polyphenol content. In this study, we elucidated the phytochemicals in ice plant (*Mesembryanthemum crystallinum*), a globally consumed vegetable, using an LC-QTOF/MS-based untargeted detection method. The phytochemicals were clustered based on their structural similarity using molecular networking and annotated using the in silico tool for network annotation propagation. To identify the bioactive compounds, eight compounds were isolated from ice plant extracts. These compounds were identified using extensive spectroscopic methods, including ^1^H and ^13^C nuclear magnetic resonance (NMR) spectroscopy. Additionally, we evaluated the antioxidant and anti-inflammatory activities of all the isolates. Among the tested compounds, three showed antioxidant activity and all eight showed anti-inflammatory activity, demonstrating the potential of ice plant as a functional food.

## 1. Introduction

The homology between food and medicine has recently been emphasized in the prevention, management, and treatment of diseases [1,2]. Foods with biological and physiological activities are referred to as functional foods. Functional foods contain various secondary metabolites and provide health benefits in addition to their basic nutritional value [3,4]. Fresh fruits and vegetables are popular among health-conscious consumers due to their polyphenol content, representing a prominent segment of functional foods [5]. Notably, the growing interest in functional foods highlights the need for comprehensive studies of their chemical composition and health benefits.

Ice plant (*Mesembryanthemum crystallinum*, Aizoaceae) is a vegetable crop consumed in India, Australia, New Zealand, and some European countries [6,7]. It is native to southern and eastern Africa and is now distributed worldwide [8]. The succulent leaves are coated with enlarged epidermal cells and provide a slightly salty taste. Recently, interest in its consumption has surged owing to its potential health benefits, such as antioxidant, anti-inflammatory, and anticancer activities [9,10]. Despite these diverse bioactive effects, limited research has been conducted on the chemical composition of ice plant, with quantitative investigations focusing on phenolic and polyphenolic compounds. However, prior knowledge of the overall chemical composition of foods is important for understanding their functional effects, and it is important to explore advanced analytical methods to uncover these chemical profiles.

Untargeted metabolomics is a useful approach for the simultaneous analysis of several compounds in plants [11]. This approach allows for comprehensive analysis of complex plant chemicals by identifying as many metabolites as possible in a given sample [12,13,14]. Advances in analytical tools have facilitated the investigation of the full chemical profile of natural products using LC-QTOF/MS spectral data. Molecular networking eases the mapping of chemical diversity by aligning each MS/MS spectrum within a dataset based on the structural similarity; the structurally-related compounds are then clustered by the MS-Cluster algorithm [12,15]. Additionally, the in silico annotation tool network annotation propagation (NAP) predicts the compound structures using a re-ranking system and enables the class-level annotation of unknown fragmented mass spectra [16]. These tools enable rapid chemical profiling of natural foods and efficient isolation of bioactive compounds through targeted separation methods.

In this study, we constructed a molecular network based on the LC-QTOF/MS spectrum of ice plant to identify its entire chemical profile. The structures of the chemical components of ice plant were predicted using NAP. Additionally, we isolated and identified the phytochemicals in the methanolic extracts of ice plant, and we evaluated their biological activities to confirm their potential application in functional food.

## 2. Materials and Methods

### 2.1. General Experimental Procedures

The HR-ESI-MS spectra were obtained using an AGILENT 6550 iFunnel Q-TOF LC-MS system. The medium pressure liquid chromatography (MPLC) used in this study was a Biotage Isolera system (Biotage, Uppsala, Sweden) equipped with ultraviolet–visible (UV-VIS) detectors. The preparative high-performance liquid chromatography (HPLC) was performed using a Jasco LC-2000Plus HPLC system (Jasco, Easton, MD, USA). The chemical shifts were reported in parts per million (ppm) from tetramethylsilane (TMS). All the nuclear magnetic resonance (NMR) spectra were recorded on a Prostar HPLC-VNMRS600-320MS spectrometer (Varian, Palo Alto, CA, USA) with methanol-*d*_4_ operated at 600 and 150 MHz for hydrogen and carbon, respectively. Data processing was performed using the MestReNova ver.12.0.1 program.

### 2.2. Sample Preparation

Ice plant samples were collected from Gyeongju, Republic of Korea. The shoots of the ice plant were dried at 50 °C for 3 days and homogenized. Thereafter, the resulting powder was sonicated twice for 3 h at 25 °C using methanol. After filtering through a 11 μm pore filter paper (Whatman, Clifton, NJ, USA), the extract was completely evaporated in vacuo and stored at −20 °C until analysis. For the LC-MS analysis, the dried samples were dissolved in LC-MS-grade methanol (JT Baker, Phillipsburg, NJ, USA) at a concentration of 1 mg/mL.

### 2.3. UPLC-QTOF-MS Analysis

The samples (1 mg/mL) were analyzed using an ultra-high-performance liquid chromatography (UPLC)-QTOF-MS system. The instrument consisted of an Agilent 1290 Infinity LC system (Agilent Technologies, Palo Alto, CA, USA) coupled with an Agilent 6550 iFunnel QTOF LC/MS system equipped with a dual Agilent Jet Stream (AJS) ESI source. The compound separation was performed using an YMC-Triart C18 column (2.0 × 150 mm, 1.9 μm; YMC Co., Sungnam, Republic of Korea) at 25 °C. The mobile phase consisted of water (0.1% formic acid) and acetonitrile (ACN, 0.1% formic acid) with the following gradient: 5–95% ACN (0–50 min), 95–100% ACN (50–50.1 min), 100% ACN (50.1–53 min), 100–5% ACN (53–53.1 min), and 5% ACN (53.1–55 min). The flow rate was 0.4 mL/min.

The MS experiments were conducted in the positive ionization mode. The QTOF settings included a mass range of 100–1000 *m*/*z* and an acquisition rate of 5 spectra/s for the MS, while for the MS/MS, the mass range was 40–1000 *m*/*z* with an acquisition rate of 3 spectra/s. The MS/MS fragmentation patterns were generated using fixed collision energies of 20 and 40 eV. The data acquisition was performed in the centroid mode using the high-resolution mode (4 GHz).

### 2.4. Phytochemical Profiling

The UPLC-QTOF-MS spectrum files of the ice plant extracts were analyzed using the molecular networking tool global natural products social molecular networking (GNPS) [17]. A molecular network was generated using the raw positive ion mode data with the following parameters: precursor ion mass tolerance of 2.0 Da, fragment ion mass tolerance of 0.5 Da, minimum pair cos of 0.7, and minimum matched fragment ion of 6. The networking results were visualized using Cytoscape 3.9.1, an open-source software package for visualizing complex networks. The molecular networking job statuses are available online at https://gnps.ucsd.edu/ProteoSAFe/status.jsp?task=db970aad9b6c4704845dae0241c351fd (accessed on: 8 December 2022).

The NAP tool from the Center for Computational Mass Spectrometry (CCMS) was used to predict the structure of the chemical components. The GNPS job (ID: db970aad9b6c4704845dae0241c351fd) was re-analyzed using the NAP_CCMS (1.2.5) workflow with the parameters set to cosine values to sub-select inside a cluster of 0.5, and the accuracy for the exact mass candidate search was 15 ppm (https://proteomics2.ucsd.edu/ProteoSAFe/index.jsp?task=4a03e82c097047278dd409538532347c; accessed on: 7 January 2024). The predicted chemical structures were classified using ClassyFire, a web-based application [18].

### 2.5. Extraction and Isolation of Ice Plant Compounds

Briefly, ice plant methanolic extract (42.4 g) was suspended in H_2_O and successively partitioned using hexane, CHCl_3_, and EtOAc to obtain EtOAc (0.5 g) and H_2_O (33.0 g) extracts after removal of the solvents in vacuo.

The H_2_O fraction was subjected to MPLC using a SNAP Ultra C18 60 g column (Biotage, Uppsala, Sweden) and eluted with 20, 40, 60, 80, and 100% methanol to yield five sub-fractions (DF1–5). The DF2 fraction was separated in an MPLC system using a SNAP Ultra C18 12 g column and eluted with a 10–90% methanol gradient to obtain two subfractions, DF2A and DF2B. Both fractions were further eluted with a 30–70% methanol gradient using MPLC (SNAP Ultra C18 12 g column), resulting in four (DF2A1, DF2A2, DF2A3, and DF2A4) and three (DF2B1, DF2B2, and DF2B3) smaller fractions, respectively. The DF2A1 fraction was subjected to HPLC and eluted with methanol:H_2_O (7:93) to yield compounds **1** (11.3 mg) and **2** (2.1 mg). Thereafter, the DF2A2 and DF2A3 fractions were eluted under the same HPLC conditions and solvent composition to yield compounds **3** (18.5 mg), **4** (1.6 mg), and **5** (2.2 mg). Compound **6** (1.8 mg) was obtained following the elution of DF2B3 with methanol:H_2_O (10:90).

The EtOAc fraction was subjected to MPLC using a Sfär C18 12 g Duo column (Biotage, Uppsala, Sweden) and eluted with a 50–90% methanol gradient to yield five fractions (EF1–5). The EF1 fraction was eluted with a 30–70% methanol gradient using the same MPLC column as the EtOAc fraction to obtain five subfractions: EF1A, EF1B, EF1C, EF1D, and EF1E. The EF1B fraction was subjected to HPLC and eluted with methanol:H_2_O (25:75) to yield compound **7** (0.6 mg), whereas the EF1C fraction produced compound **8** (1.0 mg) when eluted with methanol:H_2_O (30:70).

### 2.6. Assessment of Antioxidant Activity

#### 2.6.1. Total Antioxidant Capacity (TAC)

The TAC was measured using the Trolox equivalent antioxidant capacity (TEAC) assay. Briefly, the TAC was assessed using the OxiTec™ Total Antioxidant Capacity Assay Kit (Biomax, Seoul, Republic of Korea) according to the manufacturer’s protocol, with Trolox as the standard for comparison. Ethanol was used as the blank control instead of the reaction buffer. The absorbance was measured at 450 nm using a microplate reader (TECAN, Mannedorf, Switzerland) after 30 min of incubation [19].

#### 2.6.2. DPPH Radical Scavenging Activity

The DPPH inhibition rate was measured using the OxiTec™ DPPH Antioxidant Assay Kit (Biomax) according to the manufacturer’s protocol. Each sample and Trolox standard were prepared at final concentrations of 20, 40, 60, 80, and 100 μg/mL. Briefly, 20 μL of each sample and Trolox standard were mixed with assay buffer (80 μL) and DPPH working solution (100 μL) in a 96-well microplate. Ethanol was used as the blank control instead of the DPPH working solution. The absorbance was measured at 517 nm using a microplate reader (TECAN) after 30 min of incubation [20].

#### 2.6.3. FRAP Assay

For the assay, the FRAP solution was prepared using 10 Mm TPTZ solution, 12 Mm ferric chloride, and 0.3 M sodium acetate buffer (pH 3.6; 1:1:10). Briefly, 20 μL of each sample and ascorbic acid (20, 40, 60, 80, and 100 μg/mL) were mixed with 180 μL of the FRAP solution in a 96-well microplate. Ethanol was used as the blank control instead of the FRAP working solution. The absorbance was measured at 593 nm using a microplate reader (TECAN) after 10 min of incubation in the dark [21].

### 2.7. Assessment of Anti-Inflammatory Activity

#### 2.7.1. Cell Culture

RAW264.7 cells were purchased from the Korean Cell Line Bank (Seoul, Republic of Korea). The cells were cultured in Dulbecco’s modified Eagle’s medium (DMEM; Gibco, Thermo Fisher Scientific, Waltham, MA, USA) containing 10% heat-inactivated fetal bovine serum (FBS; Gibco, Thermo Fisher Scientific) and 1% penicillin/streptomycin (Thermo Fisher Scientific). The cells were incubated in a humidified incubator (Vision Science, Daegu, Republic of Korea) at 37 °C in a 5% CO_2_ atmosphere.

#### 2.7.2. Cell Viability Assay

The cytotoxic effect of the ice plant extracts on the RAW264.7 cells was evaluated using the CCK-8 assay (Dojindo Molecular Technologies Inc., Kumamoto, Japan). Briefly, the RAW264.7 cells were cultured in 48-well plates (1 × 10^5^ cells/wall) and pretreated with various concentrations of ice plant extracts (50, 100, and 200 μg/mL) for 1 h, followed by stimulation with LPS (1 μg/mL) for 22 h. Thereafter, 400 μL of CCK-8 working solution was added to each well and incubated at 37 °C for 90 min. The cell viability was measured using a CCK-8 solution at 450 nm using a microplate reader (TECAN) [22].

#### 2.7.3. Nitric Oxide (NO) Production

The RAW264.7 macrophages (1 × 10^5^ cells/wall) were pre-incubated in 48-well plates for 24 h and stimulated with the ice plant extracts (20, 50, and 100 μg/mL) for 24 h. The culture supernatant was collected for the quantification of NO production. The nitrite accumulation was measured by mixing 50 µL of the supernatant with 100 µL of Griess reagent (Promega, Madison, WI, USA), followed by incubation at room temperature for 10 min. The absorbance was measured at 540 nm using a microplate reader (TECAN). The NO production was determined by comparing with a dilution of the sodium nitrite standard [23].

### 2.8. Statistical Analysis

All the data analyses were performed using GraphPad Prism Version 9.0 (GraphPad, La Jolla, CA, USA). Data are expressed as the mean ± standard deviation (SD) of the values from three independent experiments. Statistical significance was determined using a one-way analysis of variance (ANOVA), followed by the Brown–Forsythe multiple comparison test. Statistical significance was set at *p* < 0.05.

## 3. Results and Discussion

### 3.1. Chemical Profiling of Ice Plant

LC-MS total ion current (TIC) chromatograms were acquired from the ice plant samples under optimized UPLC-QTOF-MS conditions for the chemical composition analysis (Figure 1). As the methanolic extract of the sample was more strongly ionized in the ESI-positive mode than in the negative mode, the positive-mode data were used for the subsequent analyses.

A molecular network was generated to investigate the phytochemical composition of ice plant using the LC-MS data (Figure 2). In total, 178 nodes and 33 clusters were detected in the 5 samples using GNPS analysis, including 1 crude methanolic extract and 4 solvent fractions (hexane, CHCl_3_, EtOAc, and H_2_O; Figure 2A). Each node represented one compound, and the nodes consisted of a pie chart based on the ratio of the peak intensities within the five samples. The thickness of the edges (widths ranging from 3.0 to 10.0) was expressed based on the cosine score of the two connected nodes, and the thicker the edge, the higher the structural similarity. Among the 178 nodes, 29 were identified using the GNPS library search tool, which compares the MS/MS spectra of unknown compounds with a library of MS/MS spectra generated from structurally characterized compounds. Additionally, the in silico annotation tool NAP successfully predicted the structure of 89 nodes among the remaining 149 unidentified nodes (Figure 2B). Among the 118 compounds annotated using GNSP and NAP, 106 were chemically classified using ClassyFire.

Additionally, the superclass, class, and subclass of each phytochemical detected in the plant were investigated and are shown as sunbursts (Figure 3). At the superclass level, the “lipid and lipid-like molecules” group had the largest proportion (69.8%), followed by “organic acids and derivatives” (8.5%), “organoheterocyclic compounds” (7.5%), “benzenoids” (7.5%), “phenylpropanoids and polyketides” (3.8%), “organic oxygen compounds” (1.9%), and “nucleosides, nucleotides, and analogues” (0.9%). Notably, prenol lipids accounted for 40.5% of the compounds in the “lipid and lipid-like molecules” group. Based on the GNPS molecular network data, lipids and lipid-like molecules, which account for most of the compounds in ice plant, were abundant in the hexane and CHCl_3_ fractions, whereas most compounds in the EtOAc and H_2_O fractions were not annotated.

Metabolomics combined with molecular networking can be used to identify the properties of a sample through the phytochemical details, especially when the chemistry is still largely unresolved. Reddy et al. [24] used molecular networking to explore the metabolomic characteristics of four plant species in the Aizoaceae family and distinguish them chemically. Molecular networking is also used as a dereplication tool for the annotation of active compounds. Gomes et al. [25] used a molecular network approach to putatively identify the active compounds in a halophyte, *Chamaecrista diphylla*, with potent antioxidant activity and isolated four active compounds.

### 3.2. Identification of Phytochemicals in Ice Plant Extract

Eight known compounds were isolated using various chromatographic and isolation approaches (Figure 1). For the structural elucidation, the isolated compounds were compared with reported ^1^H and ^13^C NMR and MS data and were identified as tyrosine (**1**) [26], uridine (**2**) [27], phenylalanine (**3**) [27], adenosine (**4**) [28], tryptophan (**5**) [26], imidazole (**6**) [29], 4-hydroxy-3-methoxybenzamide (**7**) [30], and ferulic acid (**8**) [31]. Their chemical structures are shown in Figure 4.

The ^1^H and ^13^C NMR data of the compounds were as follows:

Tyrosine (**1**): amorphous solid (11.3 mg), ^1^H NMR (600 MHz, D_2_O): *δ*_H_ 8.47 (s, 1H), 7.22–7.17 (m, 2H), 6.93–6.88 (m, 2H), 3.94 (dd, *J* = 7.5, 5.0 Hz, 1H), 3.21 (dd, *J* = 14.5, 5.0 Hz, 1H), 3.06 (dd, *J* = 14.5, 8.0 Hz, 1H). ^13^C NMR (150 MHz, D_2_O): *δ*_C_ 175.7, 156.4, 132.2, 128.3, 117.3, 57.6, 37.0.

Uridine (**2**): amorphous solid (2.1 mg), ^1^H NMR (600 MHz, D_2_O): *δ*_H_ 8.01 (d, *J* = 8.0 Hz, 1H), 5.90 (d, *J* = 4.5 Hz, 1H), 5.69 (d, *J* = 8.0 Hz, 1H), 4.19–4.17 (m, 1H), 4.15 (dd, *J* = 5.5, 4.5 Hz, 1H), 4.00 (dt, *J* = 4.5, 3.0 Hz, 1H), 3.84 (dd, *J* = 12.0, 2.5 Hz, 1H), 3.73 (dd, *J* = 12.0, 3.0 Hz, 1H). ^13^C NMR (150 MHz, D_2_O): *δ*_C_ 166.3, 152.6, 142.8, 102.7, 90.8, 86.5, 75.8, 71.4, 62.4.

Phenylalanine (**3**): amorphous solid (18.5 mg), ^1^H NMR (600 MHz, D_2_O): *δ*_H_ 7.44 (dd, *J* = 8.0, 6.5 Hz, 2H), 7.41–7.36 (m, 1H), 7.36–7.32 (m, 2H), 4.00 (dd, *J* = 8.0, 5.0 Hz, 1H), 3.33–3.26 (m, 1H), 3.13 (dd, *J* = 14.5, 8.0 Hz, 1H). ^13^C NMR (150 MHz, D_2_O): *δ*_C_ 175.4, 136.5, 130.8, 130.6, 129.1, 57.5, 37.8.

Adenosine (**4**): amorphous solid (1.6 mg), ^1^H NMR (600 MHz, D_2_O): *δ*_H_ 8.33 (s, 1H), 8.24 (s, 1H), 6.07 (d, *J* = 6.0 Hz, 1H), 4.44 (dd, *J* = 5.0, 3.5 Hz, 1H), 4.31 (q, *J* = 3.0 Hz, 1H), 3.93 (dd, *J* = 13.0, 3.0 Hz, 1H), 3.85 (dd, *J* = 13.0, 3.5 Hz, 2H).

Tryptophan (**5**): amorphous solid (2.2 mg), ^1^H NMR (600 MHz, D_2_O): *δ*_H_ 7.75 (d, *J* = 8.0 Hz, 1H), 7.55 (d, *J* = 8.0 Hz, 1H), 7.33 (s, 1H), 7.29 (ddd, *J* = 8.0, 7.0, 1.0 Hz, 1H), 7.25–7.18 (m, 1H), 4.07 (dd, *J* = 8.0, 5.0 Hz, 1H), 3.50 (dd, *J* = 15.5, 5.0 Hz, 1H), 3.32 (dd, *J* = 15.5, 8.0 Hz, 1H). ^13^C NMR (150 MHz, D_2_O): *δ*_C_ 175.9, 137.8, 128.1, 126.5, 123.6, 120.9, 119.9, 113.4, 108.9, 56.5, 27.8.

Imidazole (**6**): amorphous solid (1.8 mg), ^1^H NMR (600 MHz, CD_3_OD): *δ*_H_ 8.58 (s, 1H), 7.45 (s, 2H). ^13^C NMR (150 MHz, D_2_O): *δ*_C_ 135.2, 120.8.

4-Hydroxy-3-methoxybenzamide (**7**): amorphous solid (32.7 mg), ^1^H NMR (600 MHz, CD_3_OD): *δ*_H_ 9.77 (s, 0H), 8.55 (s, 1H), 7.57 (s, 1H), 7.51 (d, *J* = 8.0 Hz, 1H), 6.79 (d, *J* = 8.0 Hz, 1H), 3.89 (s, 3H). ^13^C NMR (150 MHz, CDCL_3_): *δ*_C_ 151.2, 148.4, 148.3, 124.8, 117.0, 115.5, 114.1, 56.4.

Ferulic acid (**8**): amorphous solid (1.0 mg), ^1^H NMR (600 MHz, CD_3_OD): *δ*_H_ 7.48 (d, *J* = 16.0 Hz, 1H), 7.15 (d, *J* = 2.0 Hz, 1H), 7.03 (dd, *J* = 8.0, 2.0 Hz, 1H), 6.79 (d, *J* = 8.0 Hz, 1H), 6.33 (d, *J* = 15.5 Hz, 1H), 3.89 (s, 3H). ^13^C NMR (150 MHz, CDCL_3_): *δ*_C_ 173.5, 150.0, 149.4, 144.7, 128.6, 123.5, 119.3, 116.5, 111.6, 56.5.

### 3.3. Antioxidant Activity

To investigate the antioxidant activity of ice plant, we assessed the TAC of the methanolic extract, four solvent fractions, and eight isolated compounds. The TAC of 1 mg of the methanolic extract, hexane, CHCl_3_, EtOAc, and H_2_O fractions was 13.37, 15.55, 17.72, 19.74, and 13.50 mM Trolox, respectively, showing that the solvent fraction had a higher TAC than the methanolic extract (Figure 5). Among the eight isolated compounds, compounds **5**, **7**, and **8** showed a higher TAC than the solvent fractions, with TAC values of 24.50, 63.43, and 60.05 mM Trolox, respectively. Particularly, compounds **7** and **8** isolated from the EtOAc fraction had a relatively higher TAC than the other compounds. Therefore, the antioxidant activities of compounds **5**, **7**, and **8** were further assessed using DPPH and FRAP assays (Table 1). The DPPH assay showed that although the three tested compounds inhibited the DPPH levels in a concentration-dependent manner, the inhibitory effect was low, even at a treatment concentration of 100 µg/mL (inhibitory percentage: 7.97, 8.52, and 21.88%, respectively). In the FRAP assay, compounds **5** and **7** also showed low FRAP values of 4.04 and 14.53 µg/mL at 100 µg/mL, respectively, whereas compound **8** exhibited a relatively high FRAP value of 71.0 µg/mL at the same concentration.

Importantly, the antioxidant activity of plant extracts is attributed to the antioxidant enzymes and secondary metabolites they contain, notably phenolic compounds such phenolic acids, flavonoids, tannins, and carotenoids [9,32,33]. Phenolic compounds primarily exert antioxidant effects through their redox properties, which involve scavenging free radicals, quenching oxygen species, and decomposing peroxides [34]. 

In the present study, we investigated the phytochemical composition of ice plant methanolic extract and solvent fractions using molecular networking based on LC-MS data (Figure 3). Our findings revealed that the “lipid and lipid-like molecules” group constituted 69.8% of the total phytochemicals, whereas the “phenylpropanoids and polyketides” group, which encompassed phenolic compounds, comprised only 3.8%. Overall, these results explain the relatively low antioxidant activity of the ice plant samples examined.

### 3.4. Anti-Inflammatory Activity

In the present study, we investigated the cytotoxicity and anti-inflammatory effects of ice plant using CCK-8 and NO production assays, respectively (Figure 6 and Figure 7). Notably, the methanolic extract and solvent fractions were not cytotoxic to RAW264.7 cells at concentrations < 100 μg/mL (Figure 6A). Additionally, the methanolic extract and four solvent fractions inhibited NO production in a concentration-dependent manner, except for the hexane and CHCl_3_ fractions (200 µg/mL), which were cytotoxic (Figure 6B). Moreover, compounds **1**–**8** were not cytotoxic to RAW264.7 cells at concentrations of 1, 5, and 10 µg/mL (Figure 7A). Furthermore, all eight compounds significantly inhibited NO production at 10 µg/mL compared with that in the positive control group, with compounds **1** and **8** inhibiting the NO levels even at a concentration of 5 µg/mL (Figure 7B). In the present study, we used ascorbic acid as a standard reagent. Ascorbic acid is well-known for its anti-inflammatory properties [35]. Notably, seven tested compounds, except compound **4**, inhibited the NO levels by 42.97–60.40% at 10 µg/mL, which was higher than the inhibition rate of ascorbic acid at 50 µg/mL (35.77%).

Research evidence indicates that halophytes possess anti-inflammatory activity [36,37,38]. For example, the shoot extracts of the Tunisian halophytic plant *Reaumuria vermiculata* inhibited NO activity, possibly due to its high phenol content [36]. Additionally, the methanol extract of *Limonium spathulatum* significantly reduced the NO concentration by 34.44% at a concentration of 25 µg/mL [37]. However, studies on the anti-inflammatory activity of ice plant are limited. Although Kang and Joo [39] assessed the anti-inflammatory activity of the ethanol extracts of each part of ice plant, to the best of our knowledge, this study is the first to confirm the anti-inflammatory activity of solvent fractions and single compounds isolated from ice plant.

In the present study, we primarily isolated single compounds from the EtOAc and H_2_O fractions, as the hexane and CHCl_3_ fractions yielded smaller masses. However, further research is needed to explore the lipid components present in the hexane and CHCl_3_ fractions. Plant steroids from medicinal plants have been utilized as anti-inflammatory agents, and oils from fatty acid-rich seeds have demonstrated anti-inflammatory properties [40,41,42]. Our findings showed that the hexane and CHCl_3_ fractions contained abundant lipid compounds (Figure 3) and possessed anti-inflammatory activity (Figure 6). Overall, this finding suggests that valuable lipid compounds can be obtained from these fractions through further isolation processes.

## 4. Conclusions

In this study, we investigated the phytochemical components of ice plant (*Mesembryanthemum crystallinum*) using an untargeted analytical method. In total, 178 compounds were detected in the methanolic extract and solvent fractions using molecular network analysis, among which 118 were annotated in silico. Lipid and lipid-like molecules accounted for the largest proportion of compounds in the ice plant extract and were abundant in the hexane and CHCl_3_ fractions. Additionally, we isolated eight compounds (**1**–**8**) from the methanolic extract and elucidated their chemical structures using ^1^H and ^13^C NMR analysis. Moreover, we evaluated the antioxidant and anti-inflammatory activities of the methanolic extract, solvent fractions, and isolated compounds. Notably, all the tested samples inhibited the NO production, and compound **8** showed significant antioxidant activity in the FRAP assay. Conclusively, our report presents a rapid, untargeted screening method for the analysis of ice plant phytochemicals and reveals the anti-inflammatory activity of the methanolic extract, solvent fractions, and eight isolated compounds, confirming the potential of ice plant as a functional food. To fully understand the health benefits of ice plant and its importance in functional food development, further studies should investigate additional bioactivities, such as its anti-cancer, anti-diabetic, and antimicrobial effects.

## Data Availability

The original contributions presented in the study are included in the article, further inquiries can be directed to the corresponding author. The original data presented in the study are openly available in GNPS at https://gnps.ucsd.edu/ProteoSAFe/status.jsp?task=db970aad9b6c4704845dae0241c351fd (accessed on: 8 December 2022) and https://proteomics2.ucsd.edu/ProteoSAFe/index.jsp?task=4a03e82c097047278dd409538532347c (accessed on: 7 January 2024).

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
