# Peer review of "LC-QTOF/MS-Based Profiling of the Phytochemicals in Ice Plant (Mesembryanthemum crystallinum) and Their Bioactivities"

_foods, 2024, doi:10.3390/foods13121820_

Round 1

Reviewer 1 Report

Comments and Suggestions for Authors

When reviewing the methodology section of your manuscript, it is crucial to ensure that all methods and procedures are appropriately referenced, especially if they are adapted or derived from previous studies or standard practices in the field. Please review this section and confirm whether there are existing studies or standard methodologies that have informed your methods. If such references exist, incorporating them into the manuscript will significantly strengthen the methodological framework.

2.6.2. DPPH Radical Scavenging Activity: It is strongly suggested that authors re confirm their findings of the antioxidant activity by measuring FRAP and/or ABTS. This will provide a solid evidence.

Major concern: authors must clearly state (in a separate paragraph) how they performed the statistical analyses of the data. Which approach did the use? parametric or non-parametric? Which confidence level?

2.7.3. Nitric Oxide (NO) Production: What is the control condition?

Authors measured NO production as an indicator of proinflammatory cytokine generation. It is suggested to measure more parameters such as measurement of interleukin (IL)-1β and tumor necrosis factor-α (TNF-α). 

Control conditions are missing. Authors must clearly mention about their controls in each experiment. 

Author Response

Reviewer 1

Comments and Suggestions for Authors

  1. When reviewing the methodology section of your manuscript, it is crucial to ensure that all methods and procedures are appropriately referenced, especially if they are adapted or derived from previous studies or standard practices in the field. Please review this section and confirm whether there are existing studies or standard methodologies that have informed your methods. If such references exist, incorporating them into the manuscript will significantly strengthen the methodological framework.

Answer: Thank you for your thorough review and insightful feedback. Accordingly, we have added the references for every standard methodology in the materials and methods section.

  1. 2.6.2. DPPH Radical Scavenging Activity: It is strongly suggested that authors re confirm their findings of the antioxidant activity by measuring FRAP and/or ABTS. This will provide a solid evidence.

Answer: We appreciate your recommendation to confirm our findings of antioxidant activity using additional assays such as FRAP and/or ABTS. According, we performed additional FRAP assay and incorporated the results into subsection 3.3. Antioxidant Activity and Table 1.

  1. Major concern: authors must clearly state (in a separate paragraph) how they performed the statistical analyses of the data. Which approach did the use? parametric or non-parametric? Which confidence level?

Answer: Thank you for your invaluable suggestion. Accordingly, we have added a separate subsection (2.8. Statistical Analysis) detailing the statistical analysis method used, including whether parametric or non-parametric approaches were employed, along with the confidence level.

  1. 2.7.3. Nitric Oxide (NO) Production: What is the control condition?

Answer: The control group in the NO experiment were cells treated with serum free medium only, and this information has been added to the legends of Figure 6 and 7.

  1. Authors measured NO production as an indicator of proinflammatory cytokine generation. It is suggested to measure more parameters such as measurement of interleukin (IL)-1β and tumor necrosis factor-α (TNF-α). 

Answer: Thank you for your invaluable suggestion that interleukin (IL)-1β and tumor necrosis factor-α (TNF-α) should be measured. However, due to the limited quantity of the isolated samples from ice plant samples, we were unable to perform these additional cytokine assays in the current study. We acknowledge the importance of these measurements and will certainly consider including them in future studies, provided that sufficient sample are available.

  1. Control conditions are missing. Authors must clearly mention about their controls in each experiment. 

Answer: Based on your comment, we have included the control conditions for each experiment in the revised manuscript. Additionally, we have updated the figure legend and table to include the control data for clarity.

Reviewer 2 Report

Comments and Suggestions for Authors

See attachment.

Comments on the Quality of English Language

None.

Author Response

Reviewer 2

The manuscript “LC-QTOF/MS-based Profiling of the Phytochemicals of Ice Plant (Mesembryanthemum crystallinum) and Their Bioactivities” by Oh and coworkes presents the use of chromatography, spectroscopy, and in silico approaches to investigate the biological activities of a methanol extract from M. crystallinum, a widely recognized medicinal plant. The use of NMR spectroscopy to identify compounds in fractions is interesting, however, this study lacks novelty since some of them have been already reported in this plant, and the bioactivities of extracts have been extensively studied in other reports. For this reason, this manuscript should be rejected from Foods. In addition, the following comments can be followed to strength this work.

Answer: Thank you for your detailed review and valuable feedback on our manuscript “LC-QTOF/MS-based Profiling of the Phytochemicals of Ice Plant (Mesembryanthemum crystallinum) and Their Bioactivities.” We appreciate your insights and suggestions to strengthen our work.

Our research places a stronger emphasis on the analytical aspects, utilizing LC-QTOF/MS combined with NMR spectroscopy to provide a comprehensive and detailed profile of the phytochemical composition. This integrative approach allows us to systematically link the chemical profile to potential bioactivities in a novel manner.

Although there have been studies on the bioactivities of M. crystallinum extracts, our study is the first to evaluate the activities of specific fractions and isolated single compounds. This is a significant step forward as it provides a deeper understanding of the specific bioactive components within the methanol extract. Additionally, although there are several reports on the antioxidant activity of ice plant, there is little literature on its anti-inflammatory activity, which was addressed in this study.

Furthermore, we also revised the text according to your comments and performed additional experiments to further improve the manuscript. We believe that our work adds valuable knowledge to the field and we respectfully request a review of the publication.

Thank you for your time and consideration.

  1. Some sections of manuscript are written in first person, change them into third person

Answer: Thank you for your comment. The subject was modified according to the reviewer's comment.

  1. Introduction is well-written but paragraphs should be connected with each other using key sentences at the end of each one.

Answer: Thank you for your invaluable suggest. Accordingly, the introduction has been revised.

  1. In L. 91, the acronym of “ACN” is not mentioned and it does not seem to be related to formic acid, revise this.

Answer: Based on your suggestion, the abbreviation of ACN (acetonitrile) has been defined in L. 94. Additionally, we confirm that 0.1% formic acid was indeed added to the acetonitrile (ACN) solution.

  1. The evaluation of the antioxidant activity of the obtained extract can be benefited if more antioxidant analyses are implemented: ABTS, FRAP, or H2O2 assays.

Answer: Based on your recommendation, we performed additional FRAP assay and incorporated the results into subsection 3.3. Antioxidant Activity and Table 1.

  1. Current evidence seems to lack the cytotoxic analysis of this plant against cancer cell lines, authors can consider including some of them to provide novel insights into the ethnopharmacology of M. crystallinum.

Answer: We appreciate your suggestion regarding the cytotoxic analysis against cancer cell lines. Unfortunately, due to the limited quantity of isolated compounds, additional experiments in this regard are not feasible at this stage. However, we acknowledge the importance of investigating the potential anticancer effects of the extracts in future studies.

  1. In addition to the previous comment, few studies regarding the bioactivities of this plant have used bacteria or fungi strains, maybe authors can revise this.

Answer: Thank you for highlighting the potential use of bacteria or fungi strains in studying the bioactivities of M. crystallinum. Regrettably, we are unable to conduct the suggested experiments due to constraints on available resources. Nevertheless, we value your suggestion and will explore this avenue in future research endeavors. Your input is greatly appreciated.

  1. In L. 166 and 173, “cells/well” are miswritten.

Answer: Thank you for keen observation. Accordingly, we have corrected it.

Reviewer 3 Report

Comments and Suggestions for Authors

The manuscript "LC-QTOF/MS-based Profiling of the Phytochemicals of Ice Plant (Mesembryanthemum crystallinum) and Their Bioactivities" refers to experimental articles, is clearly and well structured. The authors studied the methanol extracts of Mesembryanthemum crystallinum, built a molecular network based on the LC-QTOF/MS spectrum to identify the chemical profile. Eight phytochemicals from methanolic extracts of ice plants were isolated and identified, and their antioxidant and anti-inflammatory activity was evaluated. It was established that three compounds exhibit antioxidant activity, and all eight exhibit anti-inflammatory activity. Research results indicate the potential of ice plants as a functional food.

There are several comments to the article:

1. Which parts of the plant were chosen for research?

2. In section 2.5, justify why a different ratio of the elution mixture of methanol: H2O for different fractions was used for HPLC?

3. In subsection 2.6.2. describe in more detail the method of determining the radical activity of the studied samples.

4. Section 3.4 should be expanded by adding more information about the research of other scientists.

This manuscript is scientifically based, contains 36 references, is notable for its relevance and novelty. The conclusions are consistent with the evidence and arguments presented. Ethics and data availability statements are adequate.

Comments on the Quality of English Language

Minor ending of English Language

Author Response

Reviewer 3

The manuscript "LC-QTOF/MS-based Profiling of the Phytochemicals of Ice Plant (Mesembryanthemum crystallinum) and Their Bioactivities" refers to experimental articles, is clearly and well structured. The authors studied the methanol extracts of Mesembryanthemum crystallinum, built a molecular network based on the LC-QTOF/MS spectrum to identify the chemical profile. Eight phytochemicals from methanolic extracts of ice plants were isolated and identified, and their antioxidant and anti-inflammatory activity was evaluated. It was established that three compounds exhibit antioxidant activity, and all eight exhibit anti-inflammatory activity. Research results indicate the potential of ice plants as a functional food.

Answer: Thank you for your positive feedback on our manuscript and we appreciate your acknowledgment of the clear and well-structured presentation of our experimental findings.

There are several comments to the article:

  1. Which parts of the plant were chosen for research?

Answer: The entire shoot of the ice plant was used in the experiment, and this information was added to subsection 2.2. Sample Preparation.

  1. In section 2.5, justify why a different ratio of the elution mixture of methanol: H2O for different fractions was used for HPLC?

Answer: Each fraction contains different single compounds, and the HPLC solvent conditions in which each compound is eluted are different depending on their polarity, so the solvent ratio can always vary during HPLC prep.

  1. In subsection 2.6.2. describe in more detail the method of determining the radical activity of the studied samples.

Answer: Based on your suggestion, details of the DPPH assay has been included in subsection 2.6.2.

  1. Section 3.4 should be expanded by adding more information about the research of other scientists.

Answer: Based on your comments, we have cited relevant studies in section 3.4 to support our findings.

Round 2

Reviewer 1 Report

Comments and Suggestions for Authors

Revision is fine. 

Reviewer 2 Report

Comments and Suggestions for Authors

Comments on the Quality of English Language

None.

Author Response

The manuscript “LC-QTOF/MS-based Profiling of the Phytochemicals of Ice Plant (Mesembryanthemum crystallinum) and Their Bioactivities” by Oh and co-workers addressed my comments. However, some experiments were missing. It would be beneficial if the authors mentioned in their conclusions the need to continue evaluating other bioactivities of this plant and their importance to understanding the health benefits of foods.

Answer: We appreciate the reviewer's insightful comment. We have revised the conclusion of our manuscript to emphasize the need for further studies to evaluate additional bioactivities of the ice plant, such as its anti-cancer, anti-diabetic, and anti-microbial activities. Thank you for your valuable feedback, which has helped us improve our manuscript.
